

# Construction of a nomogram for predicting catheter-related bladder discomfort in patients with end-stage renal disease after renal transplantation: a retrospective study

Kao Liu, Shengli Liu, Zhiguo Peng, Na Li and Huaibin Sun

Department of Organ Transplantation, Qilu Hospital of Shandong University, Jinan, China

## ABSTRACT

**Background.** The incidence of catheter-related bladder discomfort (CRBD) is relatively high in the end-stage renal disease (ESRD) patients who underwent renal transplantation (RT). This study was designed to establish a nomogram for predicting CRBD after RT among ESRD patients.

**Methods.** In this retrospective study, we collected 269 ESRD patients who underwent RT between September 2019 and August 2023 in our hospital. The patients were divided into training set ($n = 215$) and test set ($n = 54$) based on a ratio of 8:2. Univariate and multivariate logistic regression analyses were utilized to identify the risk factors associated with CRBD after RT, and then a nomogram model was constructed. Receiver operating characteristic (ROC) and calibration curve were used to evaluate the predicting efficiency of the established nomogram.

**Results.** Multivariate logistic regression analysis showed that aberrant body mass index (BMI) (underweight: OR = 5.25; 95% CI [1.25–22.15], $P = 0.024$; overweight: OR = 2.75; 95% CI [1.17–6.49], $P = 0.021$), anuria (OR = 2.86; 95% CI [1.33–5.88]) and application of double J (DJ) stent with a diameter of >5Fr (OR = 15.88; 95% CI [6.47–39.01], $P < 0.001$) were independent risk factors for CRBD after RT. In contrast, sufentanil utilization (>100 µg) [OR = 0.39; 95% CI [0.17–0.88], $P = 0.023$] was associated with decreased incidence of CRBD. A nomogram was then established based on these parameters for predicting the occurrence of CRBD after RT. Area under the ROC curve (AUC) values and calibration curves confirmed the prediction efficiency of the nomogram.

**Conclusion.** A nomogram was established for predicting CRBD after RT in ESRD patients, which showed good prediction efficiency based on AUC and calibration curves.

# INTRODUCTION

End-stage renal disease (ESRD) patients are usually in a state of anuria due to long-term dialysis, with an extremely low urine production of less than 100 ml per day (*Choi et al., 2015*). Consequently, their bladder volume and compliance shows significant

Corresponding author
Huaibin Sun, sun-huaibin@email.sdu.edu.cn

decrease. After renal transplantation (RT) that has been considered as the optimal method for treating ESRD (*Abecassis et al., 2008*), RT recipients often complain of an aberrant increase in urine production (*Lebadi et al., 2017*) and subsequent increase in the bladder pressure, leading to urinary retention or even vesicoureteral reflux.

Urinary catheterization has been widely adopted for treating urinary retention. Inevitably, an indwelling urinary catheter may induce postoperative bladder discomfort in about 47%–90% of the patients (*Bai et al., 2015*; *Binhas et al., 2011*), which is called catheter-related bladder discomfort (CRBD) featured by urinary urgency, frequency or pain in the supra-pubic region.

According to our clinical experiences, a large number of ESRD patients would present CRBD after RT. However, rare studies have focused on the risk factors of CRBD after RT. To date, male and an indwelling urinary catheter diameter of ≥18Fr have been considered as the two major risk factors for CRBD (*Binhas et al., 2011*). CRBD would cause patient discomfort and inhibit early ambulation (*Bezherano & Kayler, 2022*). Therefore, early prediction and management of CRBD would promote the treatment outcome after RT in ESRD patients. This study was designed to establish a nomogram for predicting its occurrence among ESRD patients after RT.

## MATERIALS AND METHODS

### Patients
We retrospectively included the patients who underwent RT between September 2019 and August 2023 in our hospital. Patients met the following criteria were included: (a) ESRD patients aged 18–60 years; (b) received urinary catheterization after RT for the first time; (c) those were conscious with good compliance before the procedures. The following patients were excluded: (a) concurrent with cognitive dysfunction or psychiatric disorders, preoperative urinary tract infection (*e.g.*, frequent urination, urgency, and dysuria; or lower abdominal tenderness, percussion pain in the kidney; with or without fever; and white blood cells ≥5 per high-power field (HPF) for men and ≥10 HPF for women); (b) with a poor compliance to catheter removal; (c) those with incomplete clinical files and not willing to participate in the study. The urinary catheter was inserted before RT procedures. The protocols were approved by the Ethical Committee of Qilu Hospital of Shandong University (Approval No.: KYLL-202208-023-1). As it was a retrospective study, the requirement to obtain informed consent from eligible patients was waived by the Ethical Committee of Qilu Hospital of Shandong University.

### Study design
The included patients were divided into training set and test set based on a ratio of 8:2. In each set, the patients were divided into CRBD group and non-CRBD group according to the occurrence of CRBD after RT. Training set was utilized to establish a nomogram for predicting CRBD, while the test set was used for the validation. The study flowchart was shown in Fig. 1.

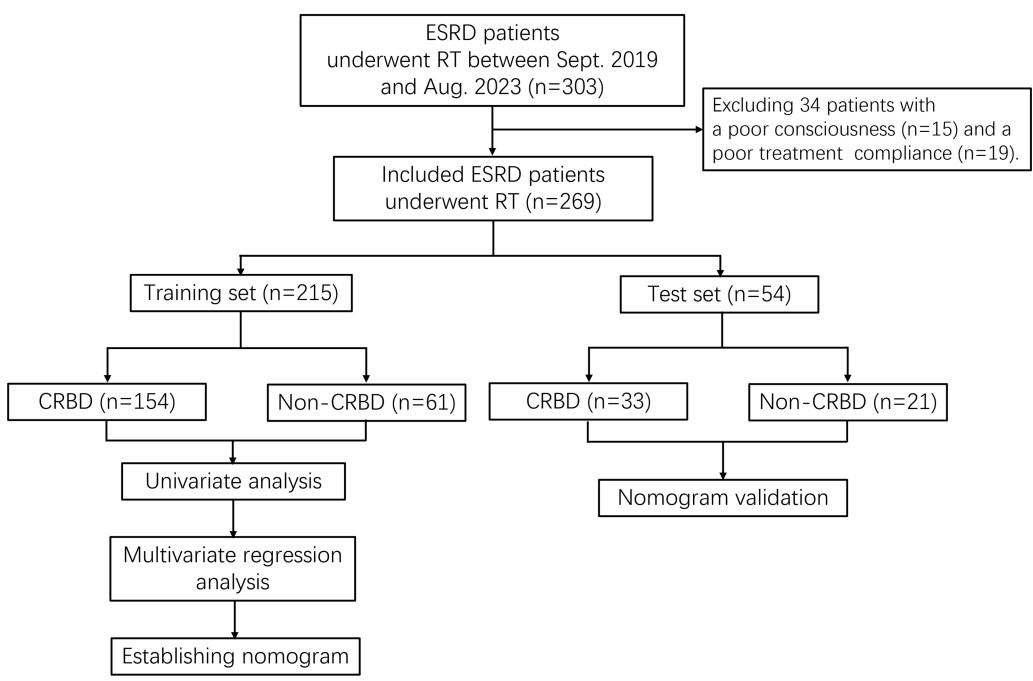

**Figure 1  Study flowchart.** ESRD, end-stage renal disease; RT, renal transplantation; CRBD, catheter-related bladder discomfort.

## Data collection

The patients included in this study were subjected to general anesthesia and nerve blockage using the standardized approach previously described by *Cho et al. (2022)*. After anesthesia, the patients were transferred to the post-anesthesia care unit for further treatment, and a nurse checked the patients' conditions to confirm whether the patient had a CRBD. CRBD was defined as a burning sensation with an urge to void or as discomfort in the suprapubic area after urinary catheters indwelling, or symptoms similar to bladder irritation (*Bai et al., 2015*; *Stojanović et al., 2015*). According to the severity, CRBD was categorized into four types, including: (a) none, reporting no CRBD even when asked; (b) mild, reporting CRBD only on questioning; (c) moderate, complained on their own without questioning but not accompanied with any behavioral responses; (d) severe, stated on their own and followed by behavioral responses such as flailing limbs, strong verbal response, or even try to pull out the urinary catheter (*Li et al., 2020*).

Demographic and clinical data were collected from each patient, including age, sex, body mass index (BMI), diarrhea, urine leakage, timing of urinary catheterization, urinary catheter type, urine production per day, diameter of utilized double J (DJ) ureteral stent, surgical time, preoperative and postoperative urine volume, dialysis method, years of dialysis, creatinine (Cr) concentration, urine red blood cell (RBC) count, patient-controlled analgesia with sufentanil (*via* pumping, for attenuating post-transplantation pain), as well as past medical history.

## Statistical analysis

Statistical analysis was performed using R software (R4.3.1 version). Normally distributed continuous variables were expressed as mean ± standard deviation and analyzed using Student's $t$-test. Variables that were not normally distributed were described as medians (interquartile ranges), and were analyzed using Mann–Whitney U test. Categorical variables were expressed as frequencies and percentages, and Chi-square test was used for the analysis. Univariate logistic regression analysis was used to compare variables between the CRBD and non-CRBD groups, and the results were provided as odds ratio (OR) and 95% confidence intervals (CI). Candidate variables with a $P$ value of <0.1 after univariate regression were adopted into Stepwise backward logistic regression, and Akaike information criterion (AIC) was used as the standard to screen modeling variables. Receiver operating characteristic (ROC) curves and calibration curves were used to evaluate the discriminative ability and consistency of the nomogram. Missing data were interpolated using random forests in machine learning using the missForest package in R language. $P < 0.05$ was considered as statistical difference.

# RESULTS

## Patients' characteristics

A total of 269 ESRD patients were included in this study, and were divided into training set (215 cases) and test set (54 cases). As shown in Table 1, there were no statistical differences between the two sets in sex, age, BMI, type and duration of dialysis, preoperative urine volume, time of urinary catheterization, dose of sufentanil, DJ stent diameter, surgical time, length of urinary catheterization, Cr concentration, as well as RBC count (all $P > 0.05$).

## Univariate regression analysis

In the training set, 154 patients (57.2%) showed CRBD after RT. Univariate regression analysis indicated that the factors associated with CRBD occurrence were time of dialysis (>12 months, OR = 2.83; 95% CI [1.26–6.32], $P < 0.001$), anuria (OR = 3.03; 95% CI [1.67–5.55], $P < 0.001$), dose of sufentanil (OR = 0.30; 95% CI [0.16–0.56], $P < 0.001$), DJ stent diameter of >5Fr (OR = 16.37; 95% CI [7.40–36.24], $P < 0.001$). In contrast, no statistical differences were observed between CRBD and the following factors, including age (OR = 1.01; 95% CI [0.98–1.03], $P = 0.501$), sex (OR = 0.80; 95% CI [0.40–1.58], $P = 0.516$), hemodialysis (OR = 2.12; 95% CI [0.98–4.55], $P = 0.055$), indwelling urinary catheter after anesthesia (OR = 1.45; 95% CI [0.79–2.66], $P = 0.23$), surgical time ($P > 0.05$), mean urinary volume on day 1 after RT (OR = 1.00; 95% CI [1.00–1.00], $P = 0.499$), time for urinary catheterization (OR = 1.73; 95% CI [0.47–6.36], $P = 0.409$), Cr concentration ($P > 0.05$), as well as urinary RBC count ($P > 0.05$) (Table 2).

## Multivariate logistic regression analysis

Factors with a $P$ value of less than 0.1 in the univariate regression analysis were included into multivariate logistic regression analysis. Multivariate logistic regression analysis showed that an aberrant BMI (underweight: OR = 5.25; 95% CI [1.25–22.15], $P = 0.024$; overweight: OR = 2.75; 95% CI [1.17–6.49], $P = 0.021$), anuria (OR = 2.86; 95% CI

**Table 1   Basic characteristics of the training set and test set.**

| Variables | Total (N = 269) | Training set (N = 215) | Test set (N = 54) | P value |
|---|---|---|---|---|
| Sex, male | 204(75.84%) | 165(76.74%) | 39(72.22%) | 0.488 |
| Age | 38.59 ± 12.59 | 38.95 ± 12.64 | 37.17 ± 12.38 | 0.352 |
| BMI | 23.33 ± 4.10 | 23.31 ± 3.99 | 23.38 ± 4.57 | 0.916 |
| Hematodialysis | 229(85.13%) | 182(84.65%) | 47(87.04%) | 0.660 |
| Duration of dialysis | | | | 0.772 |
| <6 months | 42(15.61%) | 33(15.35%) | 9(16.67%) | |
| 6–12 months | 76(28.25%) | 59(27.44%) | 17(31.48%) | |
| ≥12 months | 151(56.13%) | 123(57.21%) | 28(51.85%) | |
| Anuria | 166(61.71%) | 133(61.86%) | 33(61.11%) | 0.919 |
| Catheter indwelling duration, post-anesthesia | 122(45.35%) | 95(44.19%) | 27(50.00%) | 0.443 |
| Sufentanil, μg | 107.30 ± 21.76 | 107.42 ± 21.37 | 106.85 ± 23.46 | 0.865 |
| DJ stent diameter >5Fr | 211(78.44%) | 170(79.07%) | 41(75.93%) | 0.616 |
| Surgical time | 2.42[2.08,2.83] | 2.42[2.08,2.75] | 2.38[2.17,2.92] | 0.384 |
| Mean urine production on day 1 after RT | 3,194.55 ± 1,468.59 | 3,245.32 ± 1,552.58 | 2,992.43 ± 1,058.64 | 0.160 |
| Catheter indwelling for ≥5 days | 258(95.91%) | 205(95.35%) | 53(98.15%) | 0.699 |
| Cr | 484.00[337.00,708.00] | 507.00[344.50,717.00] | 423.00[271.00,582.00] | 0.077 |
| Urinary RBC | 8,398.20[2,505.30,27,166.10] | 9,021.20[2,456.10,27,627.75] | 6,182.50[2,671.90,18,975.90] | 0.66 |

**Notes.**

BMI, body mass index; DJ, double J; RT, renal transplantation; Cr, creatinine when carrying a catheter; RBC, red blood cell.

[1.33–5.88], $P = 0.007$) and application of DJ stent with a diameter of >5Fr (OR = 15.88; 95% CI [6.47–39.01], $P < 0.001$) were independent risk factors for CRBD (Table 3). In contrast, sufentanil utilization (>100 μg) was associated with decreased incidence of CRBD (OR = 0.39; 95% CI [0.17–0.88], $P = 0.023$).

### Establishing a nomogram for predicting CRBD

A nomogram was established based on BMI, anuria, sufentanil dose and DJ stent diameter for predicting the CRBD among the ESRD patients who underwent RT (Fig. 2). The AUC values in the training set and test set indicated a good discrimination of the nomogram in predicting CRBD (Fig. 3A). The calibration curve showed a good consistency between predicted CRBD incidence and the actual incidence (Figs. 3B and 3C).

## DISCUSSION

The incidence of CRBD in ESRD patients who underwent RT is relatively high, which may hamper the treatment outcome and compliance. In this study, aberrant BMI, anuria and DJ ureteral stent with a diameter of >5Fr were independent risk factors for CRBD, while sufentanil utilization (>100 μg) was associated with decreased incidence of CRBD. Our established nomogram based on these parameters showed a good discrimination in CRBD, presenting good consistency with the clinical experiences.

**Table 2  Univariate regression analysis for CRBD and non-CRBD groups in the training set.**

| Variables | Non-CRBD | CRBD | Univariate logistic regression | |
| --- | --- | --- | --- | --- |
| | N=61 | N=154 | OR[95%CI] | P value |
| Sex, female | 16(26.2) | 34(22.1) | 0.80[0.40, 1.58] | 0.516 |
| Age | 38.0 ± 11.4 | 39.3 ± 13.1 | 1.01[0.98, 1.03] | 0.501 |
| BMI | | | | |
|    Normal | 41(67.2) | 76(49.4) | – | – |
|    Underweight | 4(6.6) | 20(13) | 2.70[0.86, 8.42] | 0.088 |
|    Overweight | 16(26.2) | 58(37.7) | 1.96[1.00, 3.83] | 0.050 |
| Hematodialysis | 47(77) | 135(87.7) | 2.12[0.98, 4.55] | 0.055 |
| Duration of dialysis | | | | |
|    <6 months | 15(24.6) | 18(11.7) | – | – |
|    6–12 months | 18(29.5) | 41(26.6) | 1.90[0.79, 4.58] | 0.154 |
|    ≥12 months | 28(45.9) | 95(61.7) | 2.83[1.26, 6.32] | 0.011 |
| Anuria | 26(42.6) | 107(69.5) | 3.03[1.67, 5.55] | <0.001 |
| Catheter indwelling duration, post-anesthesia | 23(37.7) | 72(46.8) | 1.45[0.79, 2.66] | 0.230 |
| Sufentanil, >100 μg | 28(45.9) | 31(20.1) | 0.30[0.16, 0.56] | <0.001 |
| DJ stent diameter >5Fr | 27(44.26) | 143(92.86) | 16.37[7.40, 36.24] | <0.001 |
| Surgical time | | | | |
|    <2 h | 7(11.5) | 27(17.5) | – | – |
|    2–3 h | 46(75.4) | 102(66.2) | 0.57[0.23, 1.42] | 0.229 |
|    ≥4 h | 8(13.1) | 25(16.2) | 0.81[0.26, 2.56] | 0.720 |
| Mean urine production on day 1 after RT | 3,131.5 ± 1,538.7 | 3,290.4 ± 1,560.7 | 1.00[1.00, 1.00] | 0.499 |
| Catheter indwelling for ≥5 days | 57(93.4) | 148(96.1) | 1.73[0.47, 6.36] | 0.409 |
| Cr, μmol/L | | | | |
|    <337 | 14(23) | 36(23.4) | – | – |
|    337–484 | 18(29.5) | 36(23.4) | 0.78[0.34, 1.80] | 0.556 |
|    484–710 | 11(18) | 44(28.6) | 1.56[0.63, 3.84] | 0.338 |
|    ≥710 | 18(29.5) | 38(24.7) | 0.82[0.36, 1.89] | 0.643 |
| Urinary RBC | | | | |
|    <2,486 | 16(26.2) | 38(24.7) | – | – |
|    2,486-8,304 | 14(23) | 34(22.1) | 1.02[0.44, 2.40] | 0.959 |
|    8,304–28,089 | 20(32.8) | 39(25.3) | 0.82[0.37, 1.82] | 0.627 |
|    ≥28,089 | 11(18) | 43(27.9) | 1.65[0.68, 3.98] | 0.269 |

**Notes.**

BMI, body mass index; DJ, double J; RT, renal transplantation; Cr, creatinine when carrying a catheter; RBC, red blood cell.

To date, rare studies have focused on the CRBD in ESRD patients who underwent RT, as most of the studies on CRBD were performed in patients who underwent conventional surgery (*Adomi et al., 2019*; *Bai et al., 2015*). Many patients were likely to show CRBD after urological surgery (*Maro et al., 2014*). ESRD patients were in a state of anuria because of long-term hematodialysis. On this basis, the bladder was in an empty state for a long time. Indeed, appropriate production of urine from kidney is an important physiological process for the human beings. For example, the urination is crucial for removing the toxins. In this study, a state of anuria before RT indicated that the bladder function was poor. Thus, the

**Table 3  Multivariate regression analysis for CRBD in the training set.**

| Variables | Non-CRBD | CRBD | Multivariate logistic regression | |
|---|---|---|---|---|
| | N = 61 | N = 154 | OR[95%CI] | P value |
| BMI | | | | |
| Underweight | 4(6.6) | 20(13) | 5.25[1.25, 22.15] | 0.024 |
| Overweight | 16(26.2) | 58(37.7) | 2.75[1.17, 6.49] | 0.021 |
| Anuria | 26(42.6) | 107(69.5) | 2.86[1.33, 5.88] | 0.007 |
| Sufentanil, >100 μg | 28(45.9) | 31(20.1) | 0.39[0.17, 0.88] | 0.023 |
| DJ stent diameter >5Fr | 27(44.26) | 143(92.86) | 15.88[6.47, 39.01] | <0.001 |

Notes.
 BMI, body mass index; DJ, double J; CRBD, catheter-related bladder discomfort.

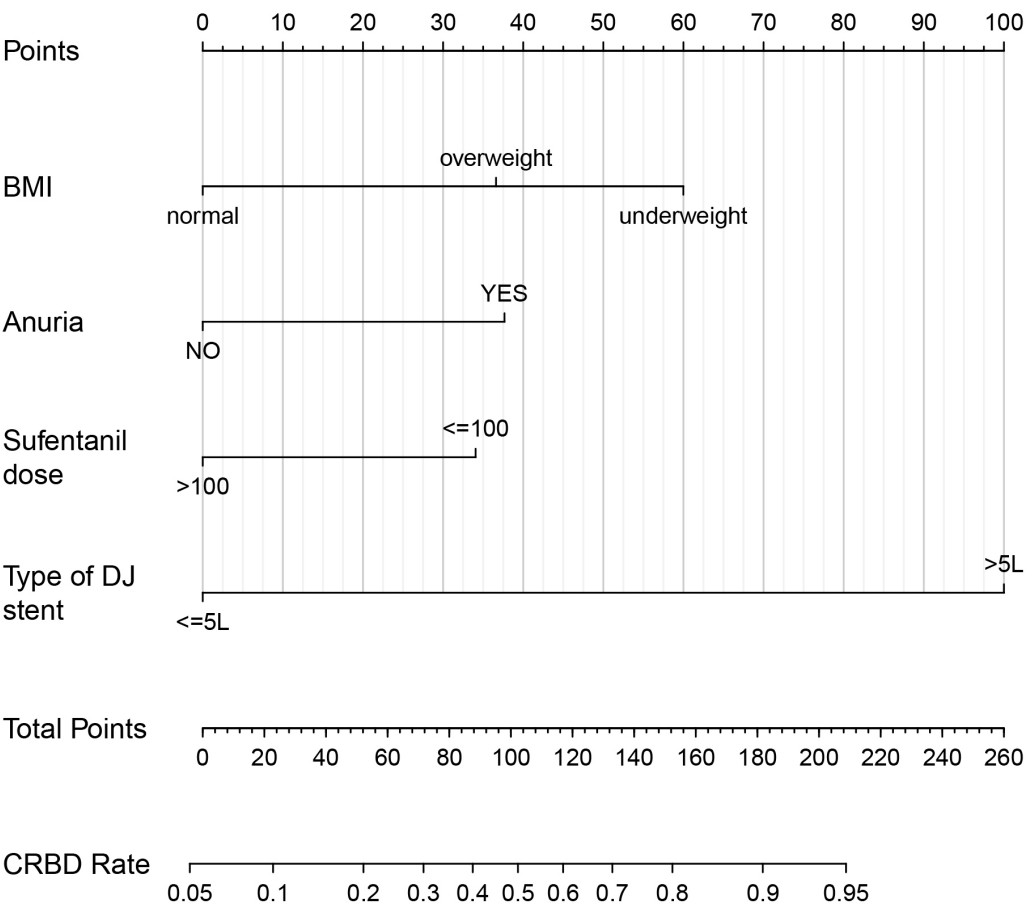

**Figure 2  Nomogram for predicting occurrence of CRBD after renal transplantation in ESRD patients.**
Anuria was defined as a urine volume of less than 100 ml per day. Sufentanil was given *via* patient-controlled intravenous pumping after renal transplantation. BMI, body mass index; DJ stent, double J stent.

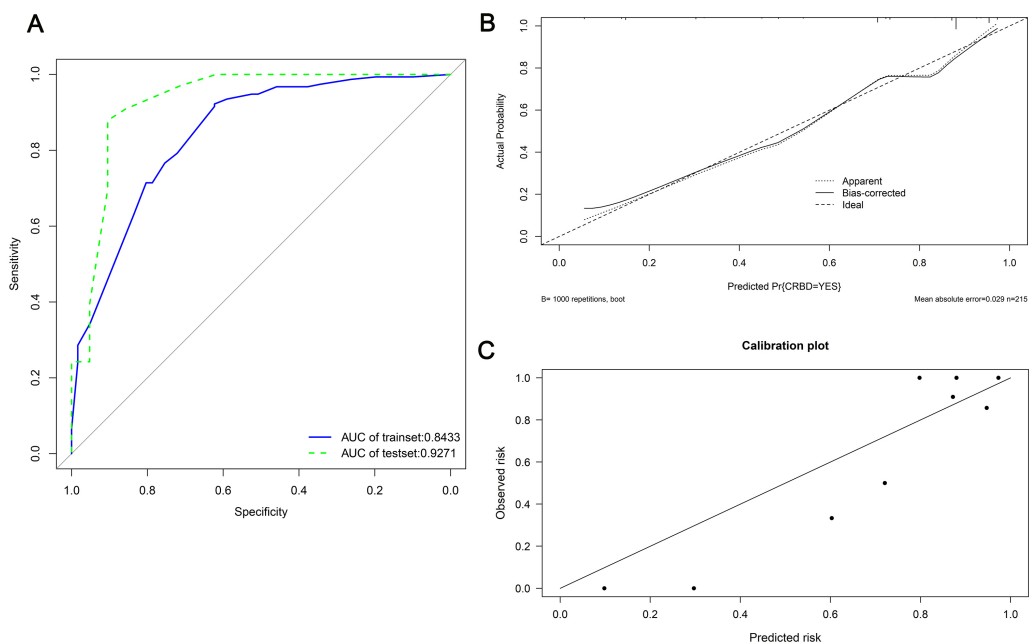

**Figure 3   ROC and calibration curves of the nomogram in both sets.** (A) ROC curves and AUC values in training set (blue) and test set (red). (B, C) Hosmer–Lemeshow calibration curves in both sets.

generated urine after RT would not be discharged from the urethra simultaneously, which required the necessity of urinary catheterization. Therefore, anuria was considered an independent risk factor for CRBD, and was adopted into the nomogram for the prediction of CRBD.

Male and an indwelling urinary catheter diameter of >18 Fr (6 mm) have been reported as the major two independent predictors of CRBD (*Mitobe et al., 2023*). Male was a risk factor for CRBD as they have a longer (18–20 cm *vs.* 4–6 cm in length) and narrower urethra than the female (*Adomi et al., 2019*; *Lim & Yoon, 2017*). Upon urinary catheterization, male patients were more likely to present CRBD due to a higher possibility of irritation (*Bach et al., 2020*; *Letica-Kriegel et al., 2019*). Inconsistently, male was not a risk factor for CRBD in our study, and the following aspects may help to explain: Our ESRD patients showed a disease duration of at least 12 months, and were in an anuria state with urine volume of less than 100 ml per day. Thus, they were in a state of bladder-empty for a long time, regardless of sex. However, both male and female encountered the same challenges in urination as the urine generation after RT was significantly higher than their previous state of anuria. All the patients previously in a state of anuria had to undergo catheterization to release the urine, which then certainly increased the risk of CRBD.

DJ ureteral stenting is a standard procedure in urological surgery (*Al-Aown et al., 2010*). Ureteral stents are mainly utilized for ureteral stabilization after surgery, with an aim for drainage through a ureter that may be obstructed, leaking, or dysfunctional (*Leslie & Sajjad, 2018*). The primary cause of DJ stent-related complications is frequently associated with leaving the stent indwelling for too long, which consequently triggers stent migration,

encrustation, as well as fragmentation. DJ ureteral stent placement had been reported to trigger hematuria, pain, and particularly CRBD (*Lim et al., 2010*). Indeed, insertion of ureteral stent would irritate the bladder trigone. Thus, the indwelling time, length and diameter were independent risk factors for the stent-related complications including CRBD (*Ilie & Ilie, 2018*; *Visser et al., 2019*). In clinical practice, attention has been paid to avoid the stent-related complications such as avoiding prolonged stent indwelling time (*Kumar et al., 2000*; *Visser et al., 2019*). In this study, DJ stent diameter of >5Fr was an independent risk factor for CRBD, as it would be more likely to irritate the urinary tract and bladder compared with the narrower and shorter stents.

The symptoms of CRBD show a large variance from burning sensation, pain in the suprapubic and penile area, to urinary urgency (*Bala et al., 2012*). Rare studies have focused on the relationship between anesthesia and CRBD after RT. In a previous study, duration of anesthesia was significantly shorter in the moderate or severe CRBD patients who did not underwent RT compared with those with none or mild CRBD (*Binhas et al., 2011*). It has been well acknowledged that patients are more likely to encounter pain after incomplete anesthesia. This indeed would trigger the pain and irritation to the patients' behaviors, which may finally result in CRBD. In this study, compared with sufentanil utilization of less than 100 $\mu$g, a dose of >100 $\mu$g would reduce the risk of CRBD, which may indicate that satisfactory anesthesia would reduce the risk of CRBD.

Our study indicated that an aberrant BMI was considered a risk factor for CRBD after RT. Some studies showed that the incidence of infection and adverse events increased in patients with obesity or overweight, compared with counterparts with a normal BMI (*Mathison, 2003*; *Olsen et al., 2008*). In a previous meta-analysis, overweight was significantly associated with increased risk of catheter-related bloodstream infection (OR = 1.43, 95% CI [1.12–1.82]) (*Wang et al., 2022*). *Al-Shaiji & Radomski (2012)* reported that patients with a BMI of 30 or more showed a higher incidence of urinary mixed leakage. In this study, an aberrant BMI (*i.e.*, overweight and underweight) was considered a risk factor for CRBD after RT, and was adopted in the nomogram. To date, our understanding on the relationship between underweight and CRBD is still limited, but we can assume that patients with underweight due to ESRD would encounter a higher possibility of immune dysfunction. Therefore, these patients are more likely to present CRBD compared with the patients with a normal BMI. In the future, we will focus on collecting more patients to analyze the relationship between BMI category and CRBD after RT.

Given the absence of practical prediction models, we established a prediction model for CRBD in ESRD patients who underwent RT. All the predictors adopted in the nomogram can be collected from the patient in an easy, rapid and routine manner. This nomogram could immediately inform the physicians and patients the risk of CRBD, and improve the clinical decision-making process, which could subsequently improve the treatment outcome and compliance.

Our study has some strengths and novelties. First, our nomogram contributed to guide the decision-making when managing patients with CRBD after RT by evaluating the risks in advance. Second, early prediction of CRBD and given appropriate management would significantly improve the quality of life among ESRD patients who underwent RT. Third,

the incidence of catheterization-related complications (*e.g.*, urinary tract infections, bladder irritation, and stone formation) might be reduced through paying attention to the risk factors.

For the limitation, this is a retrospective study which may inevitably bring in bias even we tried our best to avoid it. Additionally, the sample size was not large. Moreover, the selected risk factors associated with CRBD in our study were hard to modify in routine clinical practice. Furthermore, this is a single-centered study, and the protocols might not be applicable in other regions. Therefore, in our future work, more studies involving large-sample sized, multicentered data are required.

## CONCLUSIONS

In summary, we established a nomogram for predicting the CRBD after RT based on BMI, DJ stent diameter, a state of anuria, and sufentanil dose. AUC and calibration curves showed that the nomogram was effective in predicting the CRBD among the ESRD patients after RT. This nomogram contributed to the early prediction and risk evaluation of CRBD in surgical patients with indwelling catheters, and improved the treatment satisfaction upon appropriate treatment given in advance.

### Funding
The authors received no funding for this work.

### Competing Interests
The authors declare there are no competing interests.

### Author Contributions
- Kao Liu analyzed the data, prepared figures and/or tables, authored or reviewed drafts of the article, and approved the final draft.
- Shengli Liu performed the experiments, authored or reviewed drafts of the article, and approved the final draft.
- Zhiguo Peng performed the experiments, authored or reviewed drafts of the article, and approved the final draft.
- Na Li performed the experiments, authored or reviewed drafts of the article, and approved the final draft.
- Huaibin Sun conceived and designed the experiments, authored or reviewed drafts of the article, and approved the final draft.

### Human Ethics
The following information was supplied relating to ethical approvals (*i.e.*, approving body and any reference numbers):

The Ethical Committee of Qilu Hospital of Shandong University
## Data Availability

The raw data is available in the Supplemental File.

## Supplemental Information

Supplemental information for this article can be found online at http://dx.doi.org/10.7717/peerj.17530#supplemental-information.

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
