# Peer review of "Construction of a nomogram for predicting catheter-related bladder discomfort in patients with end-stage renal disease after renal transplantation: a retrospective study"

_PeerJ, doi:10.7717/peerj.17530_

## Round 0.1 · original submission · Major Revisions

More details about the statistical analysis should be provided and the discussions and Conclusions should be focused on the authors' own results.

**Language Note:** The review process has identified that the English language must be improved. PeerJ can provide language editing services - please contact us at [email protected] for pricing (be sure to provide your manuscript number and title). Alternatively, you should make your own arrangements to improve the language quality and provide details in your response letter. – PeerJ Staff

Reviewer 1 ·

Basic reporting

The manuscript needs English revision.
The authors need to include references in the methodology and discussion sections in order to sustain their affirmations (see comments bellow).

Experimental design

CRBD needs to be well defined!

Validity of the findings

Conclusions are not well formulated; the impact on clinical practice is not well defined.

Additional comments

1. Please clearly mention in the abstract the factors associated with catheter-related bladder discomfort (CRBD) identified in your study.
2. How was preoperative urinary tract infection defined (i.e., as an exclusion criterion)?
3. The following phrase from the methodology section is very vague and lacks reference: "Patients included in this study were subjected to general anesthesia and nerve blockade using a standardized approach according to our study protocols."
4. Provide a reference for the CRBD assessment: "The diagnosis of CRBD was based solely on the presence of symptoms, including an overactive bladder (OAB), accompanied by behavioral responses such as strong vocal reactions, flailing limbs, and attempts to remove the urinary catheter."
5. Clearly define CRBD in the methodology section and provide a reference for this definition. Since this is the study's objective, it should be stated clearly. In the current manuscript, CRBD seems more subjective than objective.
6. Explain how BMI was used in the multivariate model: "Aberrant BMI (i.e., underweight and overweight) (underweight: OR = 5.25; 95% CI [1.25, 22.15], P = 0.024; overweight: OR = 2.75; 95% CI [1.17, 6.49], P = 0.021)."
7. Discuss how the study's results are clinically relevant. Most risk factors associated with CRBD in your study are hard to modify in routine clinical practice.
8. I suggest toning down this statement: "CRBD...which is considered a significant threat to the outcome of renal transplantation."
9. Present the factors associated with CRBD in the first paragraph of the discussion.
10. Replace "gender" with "sex" throughout the manuscript.
11. Please explain how CRBD can impact the outcome of kidney transplantation, with references.
12. Describe your approach to CRBD. What is the management of CRBD in your institution compared to other centers.
13. The limitations section should be expanded. This was a single-center study; the protocols applied might not be applicable in other regions
14. The conclusion must be reformulated; it includes details that should not be present.

·

Basic reporting

The manuscript is written using clear, unambiguous, and technically correct text. In addition, adheres to professional standards of courtesy and expression. On the other hand, substantial revisions are necessary for the English language and the paper's structure could benefit from enhancements to improve clarity, especially given the complexity of certain sentences. Breaking down these sentences would facilitate a better understanding of the intricate details of the findings, particularly for the diverse audience comprising clinicians and researchers. I strongly recommend a professional proofread to address potential typographical errors and further refine sentence structures. For example, the passages between lines 28 and 28, 66 and 68, 146 and 160, and 199 and 200 require revision for English language and grammar.

Experimental design

The study is commendable for its thorough investigation of a relevant research question and its overall rigorous approach. However, there are notable areas in the methodology that could benefit from further clarification and improvement:
1. The authors should explicitly outline potential biases, such as selection bias or recall bias, and discuss how they controlled for these biases during data collection and analysis.
2. The authors should elaborate on the considerations that led to the chosen criteria, providing a clear rationale for each criterion. This will strengthen the methodological foundation and improve the study's replicability.
3. Consideration of objective measures or standardized diagnostic criteria for CRBD, along with an acknowledgment of potential subjectivity in symptom reporting, would enhance the study's validity.
4. The authors should provide a thorough discussion of the steps taken to protect patient privacy, ensure confidentiality, and justify the waiver of informed consent. Addressing these ethical aspects in detail would strengthen the study's ethical standing.
5. A detailed explanation of why certain variables were chosen for the nomogram and how their weights were determined would contribute to the transparency and interpretability of the model.
6. The authors should discuss the external validity of the nomogram, addressing its applicability to diverse patient populations and clinical settings. This will strengthen the study's relevance beyond the specific cohort studied.
7. The authors should explicitly describe how missing data were handled and detail the assumptions underlying the statistical analyses. This will contribute to the reproducibility and reliability of the study.
Addressing these suggested improvements will contribute to enhancing the overall transparency, reliability, and interpretability of the study's methodology.

Validity of the findings

The study's findings align well with its objectives and are generally valid; however, certain improvements could enhance the overall validity:
1. The study focuses on a specific hospital and timeframe. To enhance generalizability, the authors should discuss how variations in patient demographics, surgical practices, and healthcare settings might impact the applicability of their findings to a broader population. This discussion would contribute to a more comprehensive understanding of the study's external validity.
2. While the manuscript acknowledges limitations and suggests future studies, there is a lack of direct discussion on the novelty and potential impact on clinical practice. Authors are encouraged to explicitly highlight the significance and novelty of their study, explaining its contributions to existing literature and discussing how meaningful replication could further advance the field. This clarification will enhance the overall impact of the research.
3. The presented data appear robust, and statistical methods are mentioned. However, more transparency regarding the statistical methods, including how missing data were handled and assumptions made during analyses, would contribute to the overall reliability of the results. Providing these details will strengthen the study's internal validity and improve its reproducibility.
4. The acknowledgment of the retrospective nature of the study is noted, but there is a lack of extensive discussion on potential biases introduced by this design. A more thorough exploration of biases, such as selection bias or recall bias, and a clear explanation of how these biases were mitigated would strengthen the study's internal validity. Discussing the steps taken to address biases will provide a more robust interpretation of the findings.
Addressing these points will not only contribute to the internal and external validity of the study but also provide a clearer understanding of the study's impact and significance in the broader clinical context.

Additional comments

1. Following ICMJE recommendations (https://www.icmje.org/icmje-recommendations.pdf), provide explicit mention of the R software used in the research along with other equipment/software. Guidance on citing R software can be found at https://intro2r.com/citing-r.html. This ensures transparency and adherence to recommended practices for software citation.
2. In Table 1, mean urine production values on day 1 after RT are presented. However, the thousand separator symbol is missing in the Total and Training set columns. Correcting this formatting issue will enhance the accuracy and readability of the data.
3. Indicate in Table 1 that the variables are presented as medians with first and third quartiles. In this context, check the statistical analysis that reports non-normal data presented in the median and interquartile range. In opposition to presented in that section, the reported information is not the one presented for these variables.
4. Enhance the clarity and readability of Tables 1 and 2 by considering adjustments such as improved column alignment, appropriate use of headers, and ensuring consistent formatting. This will facilitate easier interpretation and understanding of the presented data.
5. To provide a more comprehensive interpretation of results, consider presenting effect sizes alongside p-values. This additional information will enhance the understanding of the practical significance of the findings and contribute to a more nuanced interpretation of the study results.

---

## Round 0.2 · Minor Revisions

The Discussion section would benefit from further revisions (in respect with applicability to different populations and regions).

·

Basic reporting

Dear Authors,

I would like to express my gratitude for the opportunity to review the enhancements made in the second version of manuscript #93410, now titled "Construction of a Nomogram for Predicting Catheter-Related Bladder Discomfort in Patients with End-Stage Renal Disease After Renal Transplantation: A Retrospective Study." The collective efforts have resulted in significant improvements to the manuscript.

I am pleased to see that all the comments and points raised in my initial feedback have been addressed in this revised version or thoroughly discussed in the response letter. The manuscript shows great promise but would benefit from some minor refinements.

First, certain sections require further editing for English language and grammar to improve clarity and readability, thereby enhancing the manuscript's overall quality for its readers.

Second, the discussion on the generalization of results could be expanded. It would be beneficial to explore how the study variables might perform across different geographic or demographic contexts. More detailed discussion on this aspect could strengthen the applicability of your findings to diverse patient populations or clinical settings.

Beyond these suggestions, I have no additional comments. Thank you again for the opportunity to contribute to the review process.

Experimental design

No comment.

Validity of the findings

No comment.

Additional comments

All comments were presented in item 1 called Basic Reporting. I don't have additional comments.

---

## Round 0.3 · accepted · Accept

All the previous suggestions were adequately resolved. No further comments.